# Spatial mapping of hepatic ER and mitochondria architecture reveals zonated remodeling in fasting and obesity

Güneş Parlakgül [1,2], Song Pang [3,6], Leonardo L. Artico [2], Nina Min [1], Erika Cagampan [1], Reyna Villa[2], Renata L. S. Goncalves[1], Grace Yankun Lee[1], C. Shan Xu [3,7], Gökhan S. Hotamışlıgil [1,4] ✉ & Ana Paula Arruda [2,5] ✉

The hepatocytes within the liver present an immense capacity to adapt to changes in nutrient availability. Here, by using high resolution volume electron microscopy, we map how hepatic subcellular spatial organization is regulated during nutritional fluctuations and as a function of liver zonation. We identify that fasting leads to remodeling of endoplasmic reticulum (ER) architecture in hepatocytes, characterized by the induction of single rough ER sheet around the mitochondria, which becomes larger and flatter. These alterations are enriched in periportal and mid-lobular hepatocytes but not in pericentral hepatocytes. Gain- and loss-of-function in vivo models demonstrate that the Ribosome receptor binding protein1 (RRBP1) is required to enable fasting-induced ER sheet-mitochondria interactions and to regulate hepatic fatty acid oxidation. Endogenous RRBP1 is enriched around periportal and mid-lobular regions of the liver. In obesity, ER-mitochondria interactions are distinct and fasting fails to induce rough ER sheet-mitochondrion interactions. These findings illustrate the importance of a regulated molecular architecture for hepatocyte metabolic flexibility.

Metabolism is an integral component of cellular function, converting nutrients into biochemical energy equivalents and building blocks for cellular growth, organismal development, and homeostasis. The evolutionary pressure to respond to fluctuations in nutrient availability has led to the development of a variety of cellular and molecular mechanisms that promote metabolic adaptation in response to acute changes in nutrient availability[1–4]. While such changes may be related to many organs, adipose and liver tissue play particularly critical roles in energy storage and mobilization, as well as programming of the diverse synthetic capacity. The liver is a central metabolic organ that exhibits vast plasticity and sharply responds to fluctuations in hormone and nutrient levels[3,4]. In the fed state, hepatocytes execute

anabolic reactions and store energy either in the form of lipids or glycogen. However, in the case of nutrient deprivation during fasting periods, hepatocyte metabolism drastically changes to produce and export glucose to other tissues and takes up lipids released by the adipose tissue to perform fatty acid oxidation and ketone body production and export. Several lines of evidence also point to heterogenous responses in sensing and disseminating metabolic stress based on the topography of hepatocytes according to their spatial localization in the liver lobe, a process called zonation[5–7]. For example, hepatocytes located near the portal vein are exposed to a higher concentration of oxygen, hormones, and nutrients and perform higher rates of gluconeogenesis and fatty acid oxidation. In contrast,

[1]Department of Molecular Metabolism and Sabri Ülker Center, Harvard T.H. Chan School of Public Health, Boston, MA, USA. [2]Department of Nutritional Sciences and Toxicology, University of California, Berkeley, Berkeley, CA, USA. [3]HHMI Janelia Research Campus, Ashburn, VA, USA. [4]Broad Institute of MIT and Harvard, Cambridge, MA, USA. [5]Chan Zuckerberg Biohub, San Francisco, CA, USA. [6]Present address: Yale School of Medicine, New Haven, CT, USA. [7]Present address: Department of Cellular & Molecular Physiology, Yale School of Medicine, New Haven, CT, USA. ✉e-mail: ghotamis@hsph.harvard.edu; aarruda@berkeley.edu

hepatocytes located near the central vein are relatively enriched in functions such as lipogenesis, VLDL, and bile acid production[5,6]. These division of tasks represents an important example of diversification and optimization of function determined by spatial localization.

Both hepatocytes' response to fasting and the regulation of their metabolic activity according to spatial distribution within the liver lobe require precise regulation of cellular processes such as nutrient sensing and response systems as well as transcriptional and translational regulation and signaling processes[8]. Proper metabolic output also demands intracellular compartmentalization in various membrane enclosed-organelles within the cell and requires communication between these compartments. Our previous studies illustrated the critical importance of a regulated subcellular architectural arrangement for metabolic homeostasis and how their disruption can lead to deterioration of liver function[9,10]. Therefore, execution of the highly dynamic and diverse metabolic program required to respond to fluctuations in nutrient availability demands subcellular structural flexibility in the liver. However, how organelles regulate and organize their architecture to respond to changes in the metabolic environment and sustain homeostasis during physiological cycles of fasting and feeding within the liver lobe are not well known. It's also poorly understood how obesity affects this dynamic process. A critical limitation to explore these questions resides in the inherit lack of precision in capturing the details of subcellular environment at high resolution in native tissue environment where the functional relevance of subcellular architectural regulation is determined.

Here, we took advantage of our recently developed workflow based on high-resolution ultrastructural imaging through enhanced FIB-SEM and machine learning-based automated image segmentation to resolve hepatocyte subcellular architecture in various nutritional states. We obtained high-resolution structural details of endoplasmic reticulum, lipid droplet, and mitochondria architecture and their interactions in entire hepatocytes in native liver tissue, from mice in fasting and fed states, across different liver zones, and in normal and obese conditions. Using this approach, we identified that fasting alters hepatic organelle morphology in a zonation-dependent manner. This dynamic structural flexibility is lost in obesity. We also show that the ER transmembrane RRBP1 is an important regulator of ER sheets' spatial organization in the liver and zonation. We conclude that the nature of ER-mitochondria interactions is differentially regulated in its continuity, distance and molecular composition in hepatocytes based on nutritional state to maintain their functional integrity.

## Results

### Fasting induces subcellular structural remodeling of hepatic ER, mitochondria, and their interactions

To examine the impact of fasting and feeding on the 3-dimensional architectural organization of hepatic organelles, we collected livers from lean male mice in fed state or fasted state following the protocol described in Supplementary Fig. 1A. To image the liver volumes at high-resolution, we utilized FIB-SEM at 8 nm isotropic resolution (in x/y/z plane). We then individually (instance-based) segmented mitochondria, ER, nucleus, and lipid droplets using automated convolutional neural network-based image segmentation, following a workflow we described earlier[10]. After reconstruction of the datasets, we identified that the total number of mitochondria per single hepatocyte varies between 2500 and 3000 in the fed state but reduced to an average of 1000 mitochondria per cell upon fasting (Fig. 1A–C and Supplementary Fig. 1B). Although fewer, the hepatocyte mitochondria in the fasted state are significantly larger, flatter (less spherical) and present higher volume and surface area compared to mitochondria from fed state (Fig. 1A, B, D–G and Supplementary Fig. 1C–F (data segregated per cell); video 1 and 2). The mitochondria shape from hepatocytes of fasted mice are strikingly more heterogenous compared to mitochondria from fed state (Supplementary Fig. 1G, H).

Specifically, the mitochondria complexity index (MCI[2])−a measure of mitochondrial branching and surface area to volume ratio[11]−is significantly higher in fasting compared to mitochondria from fed state (Fig. 1H). For example, 24.58% of mitochondria in the fasted state have an MCI[2] greater than 5 and a volume exceeding $3\,\mu m^3$, in contrast to only 2.06% in the fed state (Fig. 1I).

Mitochondria are known to make extensive contact sites with lipid droplets and these interactions regulate both mitochondrial oxidation and lipid synthesis[12,13]. Our large-scale FIB-SEM analysis show that mitochondria-LD contacts are significantly more frequent in livers from fasted compared with livers from fed mice. In some cases, a mitochondrion adapts its shape to even engulf the LD (Fig. 1J and video 3). All the morphological analyses of the organelles described above were conducted across 5−6 distinct cells for each dataset (Supplementary Fig. 1C−F).

In addition to mitochondria, fasting significantly impacts the 3D organization of the ER network. As we showed previously[10], in mid-lobular hepatocytes from fed mice, ER sheets are often organized as parallel stacks (Fig. 2A). However, upon fasting, a significant portion of rough ER sheets are remodeled as single curved sheets, often surrounding the entire mitochondria volume, forming a membrane scaffold for these organelles (Fig. 2B and video 4−6). Quantification of TEM images, using the algorithm described[10], illustrates that fasting significantly decreases the parallel organization of ER sheets (Fig. 2C). Fasting also increases smooth tubular ER abundance, although our analysis did not reach significance for this measurement (Fig. 2D).

We then mined our datasets to quantify how ER and mitochondria interact in the 3D space in hepatocytes at fed and fasted conditions. ER and mitochondria form extensive physical interactions with each other through smooth (tubular) ER-mitochondria contact sites, known as MAMs[14–17]. The MAMs have been reported to play a role in the transfer of hydrophobic molecules such as phospholipids, cholesterol, and ceramides between organelles, and to facilitate ion exchange[16–18]. More recently, interactions between rough ER and mitochondria have been described both in cell lines[19] and liver[20]. These interactions are characterized by a larger distance between the organelles and by the presence of ribosomes in their interface[19,20]. Although it's been proposed that these types of interactions are important for lipoprotein assembly[20], the understanding of the functional significance of rough ER-mitochondria interactions in physiology and disease is at its early stages. Large-scale structural and functional studies are required to establish the regulated nature of these interactions and functional consequences in physiological and pathological states.

The precise quantification of ER-mitochondria interactions in 3D, in the entirety of the hepatocytes, requires high-resolution imaging and high-throughput image analysis to address the distance and topology of ER-mitochondria at a given state of the tissue. Cryo-electron microscopy and TEM studies show that the average distance between smooth ER-mitochondria ranges from 10-25 nm while the distance between the rough ER-mitochondria interactions is ~25−80 nm with an average of 50 nm[14,20,21]. We confirmed these observations in our TEM images (Fig. 2E, F). To quantify ER-mitochondria interactions in 3D in our FIB-SEM datasets, we individually segmented mitochondria and generated objects consisting of the most outer voxel of the mitochondria membrane at a single voxel (8 nm) thickness (mitochondria surface). We then expanded the ER-annotated voxels at different distances based on the known distances of MAMs and rough ER-mitochondria described above. For the MAMs, we expanded the ER-annotated voxels by 24 nm (3 voxels) and marked the ER interaction surface on the mitochondria surface (red in Fig. 2G). For the quantification of the ER sheet-mitochondria interaction, we expanded the ER-annotated voxels by 56 nm (7 voxels) and marked the ER interaction surface on the mitochondria surface (purple in Fig. 2G). Any overlap that was marked by the 24 nm expanded ER-annotated voxels was not included in the measurements of interactions at

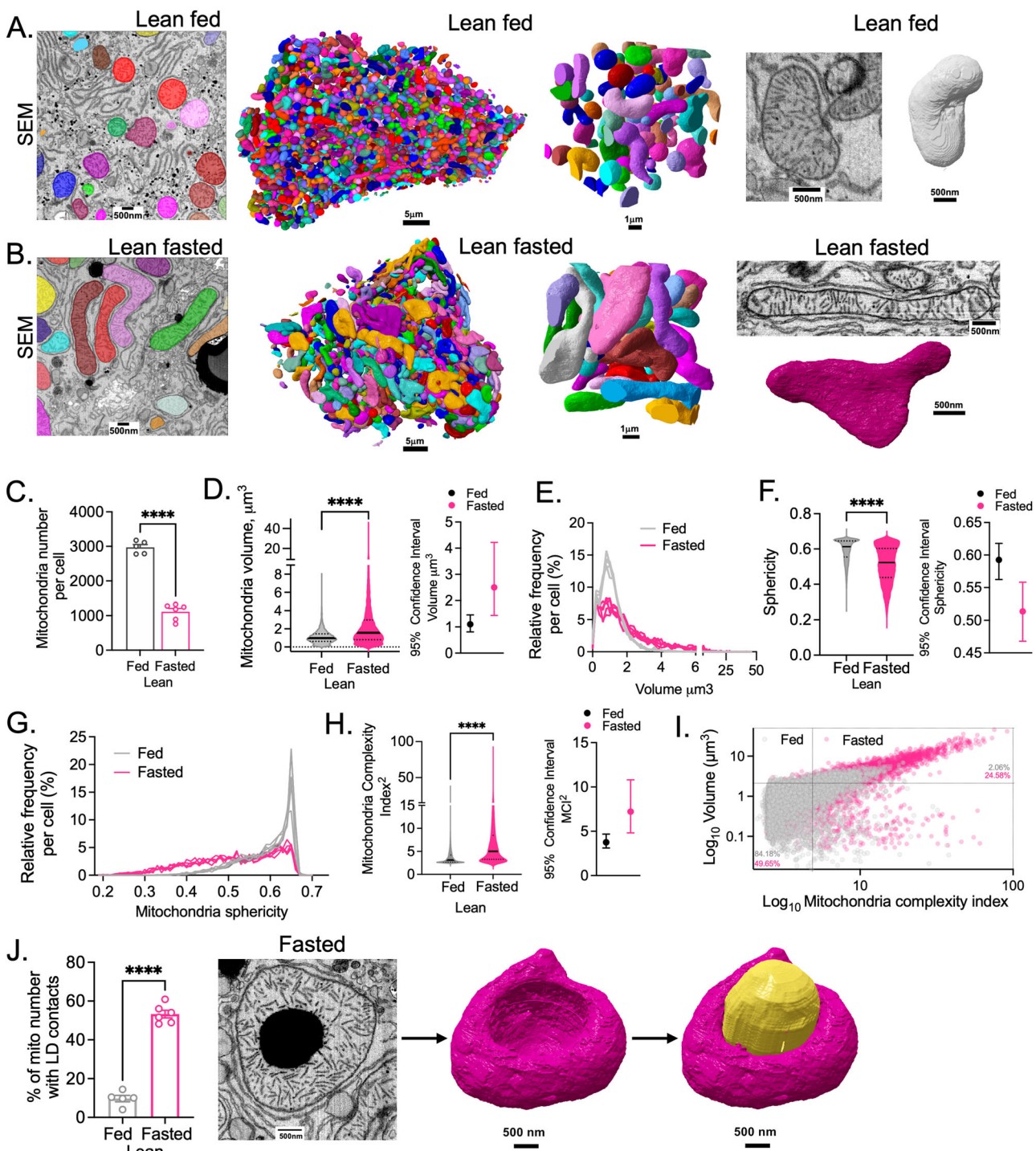

**Fig. 1 | Impact of feeding and fasting on the mitochondria 3D architecture in liver tissue. A** SEM images, 3D segmentation and reconstruction of individual mitochondria in mid-lobular hepatocytes in fed state. Scale bars from left to right: 500 nm, 5 μm, 1 μm, 500 nm, 500 nm. **B** SEM images, 3D segmentation and reconstruction of individual mitochondria in mid-lobular hepatocytes in fasted state. Scale bars from left to right: 500 nm, 5 μm, 1 μm, 500 nm, 500 nm. **C** Number of mitochondria from individual hepatocyte volumes in fed ($n = 5$ cells) and fasted ($n = 6$ cells) state. **D** Left: Mitochondria volume; Right: 95% of confidence interval for data in **D**. **E** Mitochondria volume frequency distribution. **F** Left: Mitochondria sphericity; Right: 95% of confidence interval for data in **F**. **G** Mitochondria sphericity frequency distribution. **H** Left: Mitochondria complexity index (MCI$^2$) calculated based on formula described[11] and in methods section; Right: 95% of confidence interval for data in **H**. **I** Bivariate plot of volume and MCI$^2$ for lean fed (gray) and lean fasted (pink). Each point represents a single mitochondrion. **J** Left: Quantification of % of mitochondria displaying interaction with lipid droplet (LD); Left: example of LD-mitochondria interaction in fasted state (left). Fed, $n = 5$ cells and fasted, $n = 6$ cells. Scale bars: 500 nm. For **D**–**I** quantifications were performed in 5 cells from fed data set and 6 cells for fasted conditions. The total mitochondria quantified were $n = 14,855$ in fed state and $n = 6689$ in fasted state. Fasted state corresponds to 20 h (overnight) food withdrawal. For the bar graphs, data are shown as mean ± s.e.m.; For **C** and **J**, two-tailed unpaired $t$-test ****$p < 0.0001$. For **D**, **F**, and **H**, two-tailed unpaired $t$-test ****$p < 0.0001$ and permutation test ****$p < 0.0001$.

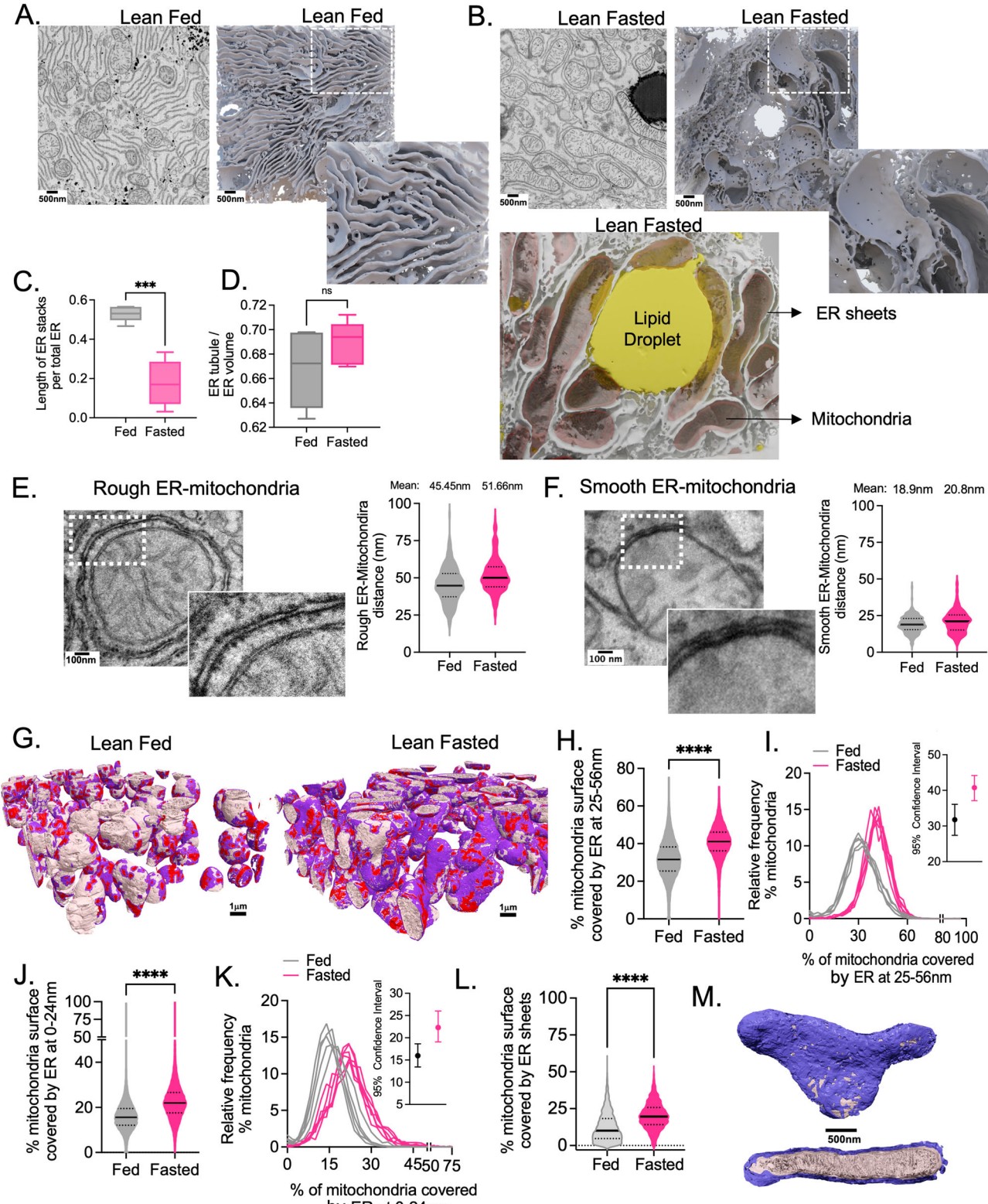

distances of 25-56 nm. We then calculated the ratio between the mitochondria surface interacting with the ER and the whole mitochondria surface (Details of the quantification pipeline are described in Supplementary Fig. 2A, B). It is important to note that while the highest resolution can be achieved with our FIB-SEM analysis, we could not detect ribosomes in the same structural background, due to technical limitations of the staining protocol in this method. However, the TEM analysis, shows that the majority of ER close to mitochondria at

25-56 nm distance contains ribosomes and is therefore formed by the rough ER.

Analysis of 14,855 (fed) vs 6678 (fasted) mitochondria show that fasting induces a significant overall increase in ER-mitochondria interactions (0-56 nm) (Supplementary Fig. 3A, B). Interestingly however, the interactions induced by fasting are predominantly formed by ER at 25-56 nm distance (Fig. 2G–I and Supplementary Fig. 3C), although there is also an augmentation of tight contact-sites between

**Fig. 2 | Impact of feeding and fasting on the endoplasmic reticulum (ER) 3D architecture and ER-mitochondria interactions in liver tissue.**
**A**, **B** Reconstruction of ER network in fed (**A**) and fasted (**B**) state ($1000 \times 1000 \times 400$ voxels, $8 \times 8 \times 3.2\ \mu m^3$). Scale bars: 500 nm. Images show SEM (left) and 3D rendering (right) of FIB-SEM data, rendered in Houdini software. In B, lower panel shows 3D rendering of ER and lipid droplet from FIB-SEM data using Houdini software. **C** Quantification of length of parallel organized ER stacks per total ER from TEM ($n = 5$ images per condition). **D** Ratio of ER tubules per total ER volume quantified from FIB-SEM data ($n = 5$ cells volumes for fed and $n = 6$ cell volumes for fasted). **E** Right: TEM image of mitochondria surrounded by rough ER. Scale bar: 100 nm. Left: quantification of distance between rough ER sheets-mitochondria membranes in TEM images. Number of mitochondria quantified $n = 248$ fed; $n = 300$ fasted. **F** Right: TEM image of mitochondria surrounded by smooth ER. Scale bar: 100 nm. Left: Quantification of distance between smooth ER-mitochondria membranes in TEM images. Number of mitochondria quantified $n = 246$ fed, $n = 183$ fasted. **G** 3D reconstruction of mitochondria volumes marked by areas of interactions with the ER at 0–24 nm distance (red) and 25–56 nm

distance (purple) in fed (left, $n = 14,855$ mitochondria) and fasted (right, $n = 6678$ mitochondria) conditions. Scale bars: 1 μm. **H, I** Quantification of the mitochondria surface covered by ER at 24–56 nm distance in fed ($n = 14,855$ mitochondria) and fasted ($n = 6678$ mitochondria) conditions. Inset: 95% of confidence interval for data in **H. J, K** Quantification of the mitochondria surface covered by ER at 0–24 nm distance in fed ($n = 14,855$ mitochondria) and fasted ($n = 6678$ mitochondria) conditions. Inset: 95% of confidence interval for data in **K. L** Quantification of % of mitochondria surface covered by ER sheets at 0-56 nm in a whole hepatocyte volume ($n = 2786$ mitochondria for fed and $n = 1275$ mitochondria for fasted). ER sheets were segmented using the previously described algorithm[10]. **M** 3D rendering of a mitochondrion from fasted cell covered by a single ER sheet (purple). Scale bar: 500 nm. For box-and-whisker plots in **C** and **D**, the line inside the box shows the median value. The bounds of the box represent the 25th–75th percentiles, with whiskers at minimum and maximum values, two-tailed unpaired $t$-test, ***$p = 0.0002$. For **H** and **J**, two-tailed unpaired $t$-test ****$p < 0.0001$ and permutation test ****$p < 0.0001$.

ER-mitochondria (at 0–24 nm), albeit at a lower rate (Fig. 2J, K and Supplementary Fig. 3D). We further refined our quantification analysis by employing a voxel-by-voxel measurement approach to quantify ER-mitochondria interactions extending up to 56 nm. Through this detailed analysis, it became evident that the most pronounced differences in the frequency distributions of ER sheet-mitochondria interactions during fasting occur at distances ranging from 49 to 56 nm (Supplementary Fig. 3E). As an additional method of quantification, we re-segmented ER segregating the ER sheets out of the total ER[10]. We then measured the interactions between ER sheets and mitochondria within 0-56 nm distance. As shown in Fig. 2L, M, fasting largely increased the interactions of ER sheets with the mitochondria. Thus, altogether, our 2D and 3D image analysis revealed that in mid-lobular hepatocytes under fasting condition, most of the ER around the mitochondria is formed by a single, continuous ER sheet covering the mitochondria volume.

To define the temporal dynamics of hepatic ER and mitochondria remodeling during fasting, we performed a time-course of food withdrawal up to 24 h and collected livers at various time points for ultrastructural analysis (Supplementary Fig. 4A). As expected, fasting induces a stepwise reduction in serum glucose levels (Supplementary Fig. 4B). TEM images of livers derived from multiple mice exposed to increased fasting times show that the increase in mitochondria size promoted by fasting is progressive over time, but it is more pronounced in mice subjected to 16-24 hours of fasting (Supplementary Fig. 4C–E). We then quantified the ER-mitochondria interactions over the fasting time course using the approach described in Supplementary Fig. 4F. Similar to mitochondria area, the increase in interactions between the rough ER and the mitochondria (at a distance between 25 and 80 nm) is more evident after 16–24 h fasting, while smooth ER-mitochondria interactions (MAMs, at a distance between 0–24 nm) increased earlier (6 h) in the time course (Supplementary Fig. 4G–J). It is important to note that along the rough ER-sheet-mitochondria interactions there are areas of tight contacts devoid of ribosomes.

### Subcellular architectural organization and response to fasting is zonated in the liver

The liver is not a homogeneous tissue; metabolic processes carried out by the hepatocytes are compartmentalized in a heterogeneous fashion across the liver lobule, a process called liver zonation[5,6]. Studies using electron microscopy have identified considerable differences in the mitochondrial structure in hepatocytes across the liver lobe[22–24]. To investigate whether fasting induced structural alterations in ER and mitochondria morphology and if their interaction differ across the distinct liver zones, we used TEM and FIB-SEM. We imaged organelle ultrastructure in hepatocytes located near the portal vein (periportal—PP), the central vein (pericentral—PC) and in the mid-lobular or

intermediary (IT) region of the liver lobe (Fig. 3A). The portal vein was identified by the presence of the portal triad comprising the hepatic artery, bile duct and portal vein and the central vein by the absence of these structures (Supplementary Fig. 5A–C). We detected that in the fed state, mitochondria from hepatocytes located in the periportal zone are significantly larger compared to mitochondria located at the pericentral and mid-lobular zones (Fig. 3B, C, E, F and video 7–8). Fasting induces mitochondria enlargement and this alteration is most prominent in mid-lobular (intermediary) hepatocytes with no significant differences found in pericentral hepatocytes (Fig. 3C, D). ER architecture also significantly differs according to hepatocyte zonation. In the fed state, the rough ER sheets are predominantly found as parallel stacks in peri-central and mid-lobular hepatocytes (Fig. 3B, G, H and video 7–8). In contrast, periportal hepatocytes present a reduction in parallel organized ER sheets and an increased abundance of single rough ER sheets in proximity with the mitochondria (Fig. 3D, G-K and video 7–8). Importantly, the effect of fasting on the reorganization the rough ER sheets around the mitochondria is also more frequent in periportal and mid-lobular hepatocytes and not detected in pericentral hepatocytes (Fig. 3D, H–K).

Altogether, these results show that in the fed state ER and mitochondria morphology vary between the different liver zones. Rough ER-mitochondria interactions are more evident in periportal hepatocytes. Importantly, hepatocyte subcellular remodeling induced by fasting is preferentially localized in mid-lobular and periportal areas. This localized response is likely a requirement for shifts in fuel selection and metabolic activity of these hepatocytes. For example, mitochondria from periportal hepatocytes have been shown to display higher fatty acid oxidation capacity compared with pericentral hepatocytes[25]. During fasting the increased demand of fatty acid oxidation likely results in the expansion of periportal phenotype to mid-lobular hepatocytes. Our data also suggests that the formation of rough ER-mitochondria interactions may be related with metabolic functions performed in periportal and midlobular hepatocytes such as fatty acid oxidation and ketone body production. Lastly, but importantly, these data highlight that when analyzing the impact of nutritional fluctuations on liver organelle morphology, it's important to take into consideration the localization of hepatocytes in the liver lobe.

### Fasting-induced dynamic remodeling of organelle architecture is compromised in obesity

Preservation of metabolic flexibility and responses to transitions in nutrient availability in tissues are critical for homeostasis and metabolic health. If a dynamic molecular architecture is a critical determinant of metabolic flexibility, it may then be postulated that pathological metabolic states, such as obesity, will present alterations in structural regulation of subcellular organelles. To determine how

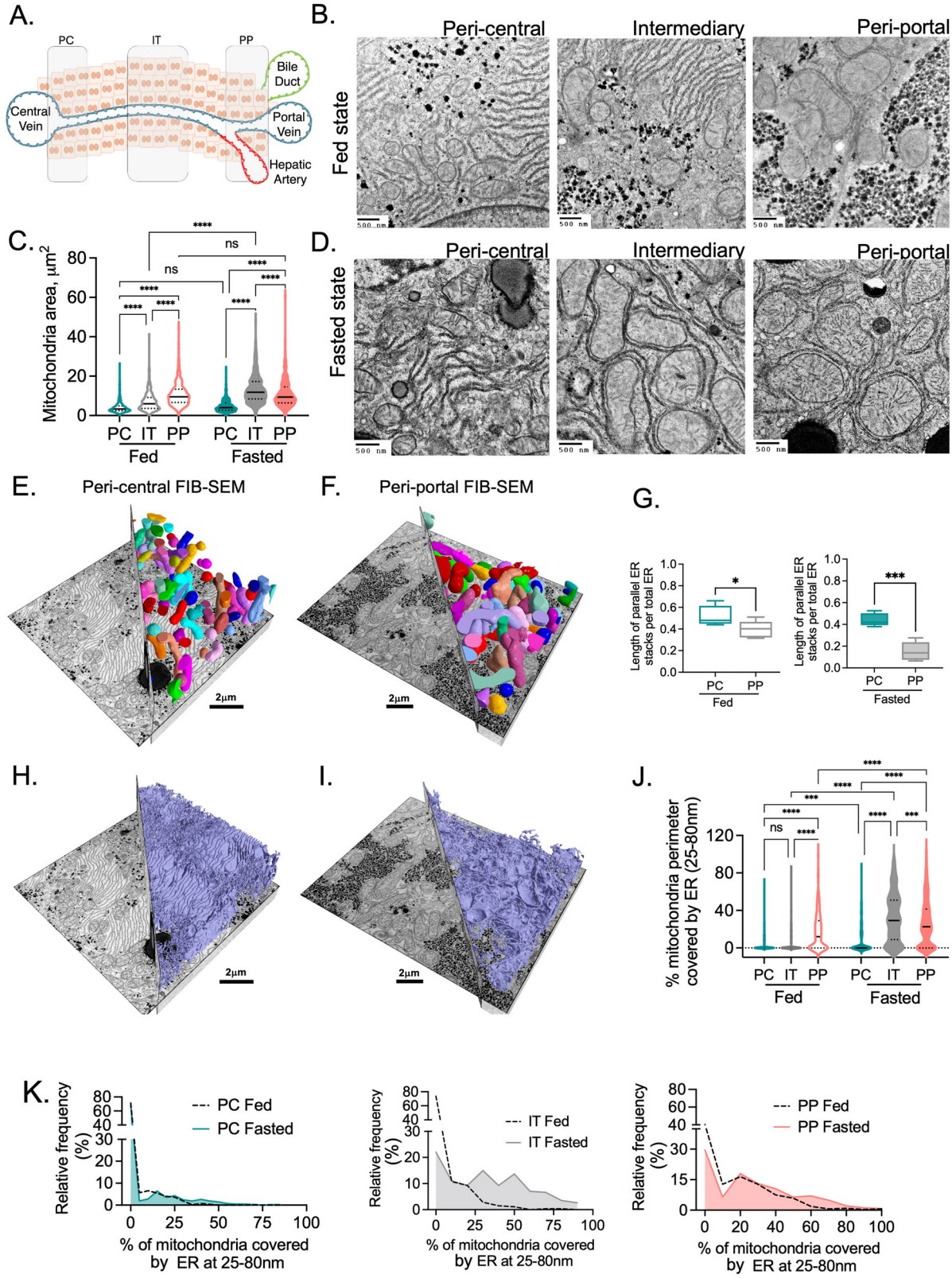

obesity influences structural dynamics of the organelles and inter-organelle interactions, we resolved the 3D ultrastructural organization of ER and mitochondria in full hepatocytes of large volumes of liver tissue from genetically obese mice in fed and fasted states and compared these to their lean counterparts.

As shown in Fig. 4A, the number of mitochondria in hepatocytes of obese mice in fed and fasted state are similar. Fasting induces a small decrease in mitochondria volume (mean: 1.23 vs 1.12 μm³, fed vs fasted, respectively) (Fig. 4B–E) and a minimal decrease mitochondria sphericity and complexity (Fig. 4F, G and Supplementary Fig. 6A). Although these data show very significant $P$ values (due to the large data points), the 95% confidence level of the data in Fig. 4D, F, show large overlap. However, in the wild-type lean controls, the extent of differences in mitochondria number, volume (mean: 1.11 vs 2.53 μm³,

**Fig. 3 | Hepatic ER and mitochondria architectural organization in fed and fasted state across the liver lobule. A** Illustration of the different hepatic zones highlighting hepatocytes in peri-central region (PC), peri-portal region (PP) and intermediary (IT) region of the liver lobule in between PC and PP. **B** Representative TEM images from peri-central, intermediary, and peri-portal regions in fed state. Scale bars: 500 nm. **C** Quantification of mitochondria area from indicated zones in fed (n = 550 for PC, n = 530 for IT, n = 350 for PP) and fasted (n = 419 for PC, n = 231 for IT, n = 524 for PP) conditions. Two-way ANOVA, Tukey's multiple comparisons test ****p < 0.0001. **D** Representative TEM images from peri-central, intermediary, and peri-portal regions in fasted state. Scale bars: 500 nm. **E, F** FIB-SEM images of mitochondria in hepatocytes located in peri-central (**E**) and peri-portal (**F**) regions of the liver from mice in fed state. Scale bars: 2 μm. **G** Quantification of length of

parallel organized ER stacks per total ER from TEM of mice in fed (left) and fasted state (right) (n = 5 images per condition). For box-and-whisker plots, the line inside the box shows the median value. The bounds of the box represent the 25th–75th percentiles, with whiskers at minimum and maximum values, Two-tailed unpaired *t*-test, *p < 0.05, ***p < 0.0003. **H, I** FIB-SEM images of ER in hepatocytes located in peri-central (**H**) and peri-portal (**I**) regions of the liver from mice in fed state. Scale bars: 2 μm. **J** Quantification of the mitochondria surface covered by ER at 25–80 nm distance in fed and fasted conditions. Fed (n = 549 for PC, n = 529 for IT, n = 322 for PP) and fasted (n = 419 for PC, n = 227 for IT, n = 513 for PP) conditions. Two-way ANOVA, Tukey's multiple comparisons test ***p < 0.0003 ****p < 0.0001. **K** Relative frequency distribution of % of mitochondria covered by ER at 25–80 nm in PC (left), IT (middle), PP (right).

fed vs fasted, respectively) and sphericity (mean: 0.59 vs 0.51, fed vs fasted, respectively) induced by fasting are higher and structural reorganization is far more pronounced (Fig. 1A–G vs Fig. 4B–G). Also, the overall number of mitochondria per cell does not change in the fed state comparing hepatocytes from lean and obese mice (2971 vs 3544 average number per cell) (Figs. 1C and 4A), although they do show higher volume (1.11 vs 1.23 μm³ average volume per cell) (Figs. 1D and 4D). The hepatic mitochondria from obese mice display extensive contacts with lipid droplets. However, these interactions are not further increased by fasting (Supplementary Fig. 6B). Interestingly, we observed that the spatial distribution of mitochondria within the cell markedly differs in hepatocytes from lean and obese mice in fed and fasted states. In lean mice in the fasted state, mitochondria are significantly closer to the nucleus compared with the fed state (Supplementary Fig. 6C, D, G–I). More strikingly, in hepatocytes from obese mice, the mitochondria distance from the nucleus is significantly higher compared to lean hepatocytes both in fed and fasted state (Supplementary Fig. 6C–I). This is likely resulting from displacement of the mitochondria from the center of the cell to the periphery due to the large accumulation of lipid droplets. Although the functional implication of these findings is unclear, there is substantial evidence showing that the proximity between mitochondria and the nucleus is important for efficiency of mitochondrial protein translation/import, retrograde responses and in fuel selection[26,27].

Next, we examined the overall architecture of the ER in obese hepatocytes in response to feeding and fasting. As we have previously shown, in the fed state, ER morphology in obese hepatocytes is characterized by an enrichment in ER tubules and decreased ER sheet/tubule ratio compared with their lean counterparts[10]. Here, we show that in fasting, the overall ER architecture in hepatocytes from obese mice does not significantly change compared to the obese hepatocytes in fed state (Supplementary Fig. 7A). Therefore, the dynamic remodeling of ER and mitochondria induced by fasting in lean mice is significantly impaired by the chronic obesity and organelle morphologies are essentially locked-in a static state. In other words, the hepatocyte subcellular structural flexibility is lost in obesity. We then addressed the impact of obesity on the regulation of ER-mitochondria interactions in the livers of mice in fed and fasted states. We and others have previously shown that in the fed state, obesity leads to enrichment of MAMs in the hepatocytes, which is implicated in excessive calcium signaling between these organelles and mitochondrial dysfunction[9,28–36]. However, decreased interactions between ER-mitochondria contact sites in obesity has also been reported[37,38]. Here, we examined ER and mitochondria interactions across different distances in more than 50,000 mitochondria in liver tissue samples from lean and obese mice in fed and fasted state. This massive analysis allowed us to eliminate potential methodological issues that may influence overall conclusions about MAM quantification in tissue related to sample size, sensitivity, and spatial resolution which may differ between approaches. As shown Supplementary Fig. 7B, C, obesity significantly increases overall (0–56 nm) ER-mitochondria connections in both fed and fasted states. When we segregated the

quantification analysis into 0–24 nm and 25–56 nm intervals, we observed that obesity consistently leads a significant increase in the MAMs at 0–24 nm (Fig. 4H–J, Supplementary Fig. 7D–E, and Video 9). Interestingly, however, fasting-induced rough ER sheet-mitochondria interactions (25–56 nm) are disrupted in obesity (Fig. 4K–M and Supplementary Fig. 7F–H).

To enhance the robustness of our findings and increase the number of biological replicates, we performed 2D TEM imaging and quantification analysis of ER-mitochondria interactions in mid-lobular hepatocytes of livers from multiple lean and obese mice in fed and fasted state. The quantification pipeline used was the same described in the Supplementary Fig. 4F. Consistent with our FIB-SEM analysis, fasting induced a significant increase in total and rough ER-mitochondria interactions in lean mice and these structural remodeling was blunted in obesity (Supplementary Fig. 8 and Supplementary Fig. 9A, B). We also confirmed that obesity leads to a marked increase in MAMs in both fed and fasted conditions (Supplementary Fig. 9C). Therefore, obesity impacts the dynamic regulation of organelles in response to fasting and leads to distinct outcomes on ER-mitochondria interactions in its continuity, distance, and frequency.

## RRBP1 is a critical regulator of rough ER sheet remodeling in fasting

The structure of the ER is determined by the relative abundance of ER membrane shaping and stabilizing proteins such as Climp-63, ribosome receptor binding protein 1 (RRBP1 or p180) or the reticulon homology domain-containing proteins[39,40]. RRBP1 has also been shown to regulate ER-mitochondria interactions in cell lines and liver hepatocytes[19,20]. However, whether RRBP1 expression and localization is modulated by fasting and obesity and is required for ER remodeling in fasting has never been tested. To address these questions, we determined the expression levels of RRBP1 in total liver lysates and ER fractions from lean and obese mice in fed and fasted states. Fasting leads to a slight increase in RRBP1 expression levels in total lysates (Fig. 5A and Supplementary Fig. 10A) which is more evident in isolated ER fractions (Supplementary Fig. 10B). Noticeably, RRBP1 immunoblot displays multiple bands varying from ~180 kDa, the expected molecular weight of RRBP1, to 90–100 kDa (Fig. 5A and Supplementary Fig. 10A, B). All the bands are specific, since they are absent in liver lysates from RRBP1-deficient mice (RRBP1-KO) (Fig. 5A). In fact, RRBP1 contains a region with tandem repeats of a decapeptide (up to 54 repeats) and has been shown to be unstable during sample preparation[41]. We also detected higher RRBP1 levels in immunostaining of tissue sections from lean mice in fasted compared to fed state (Fig. 5B, C). Interestingly, these immunostaining experiments revealed that RRBP1 levels are not only higher in fasted livers, but also that its expression is zonated, being preferentially localized to midlobular and periportal regions of the liver (Fig. 5B, C). Consistent with this observation, mining of published spatial proteomic data across liver zones[42], revealed that RRBP1 expression is higher in periportal hepatocytes (Supplementary Fig. 10C). Thus, RRBP1's localization within liver

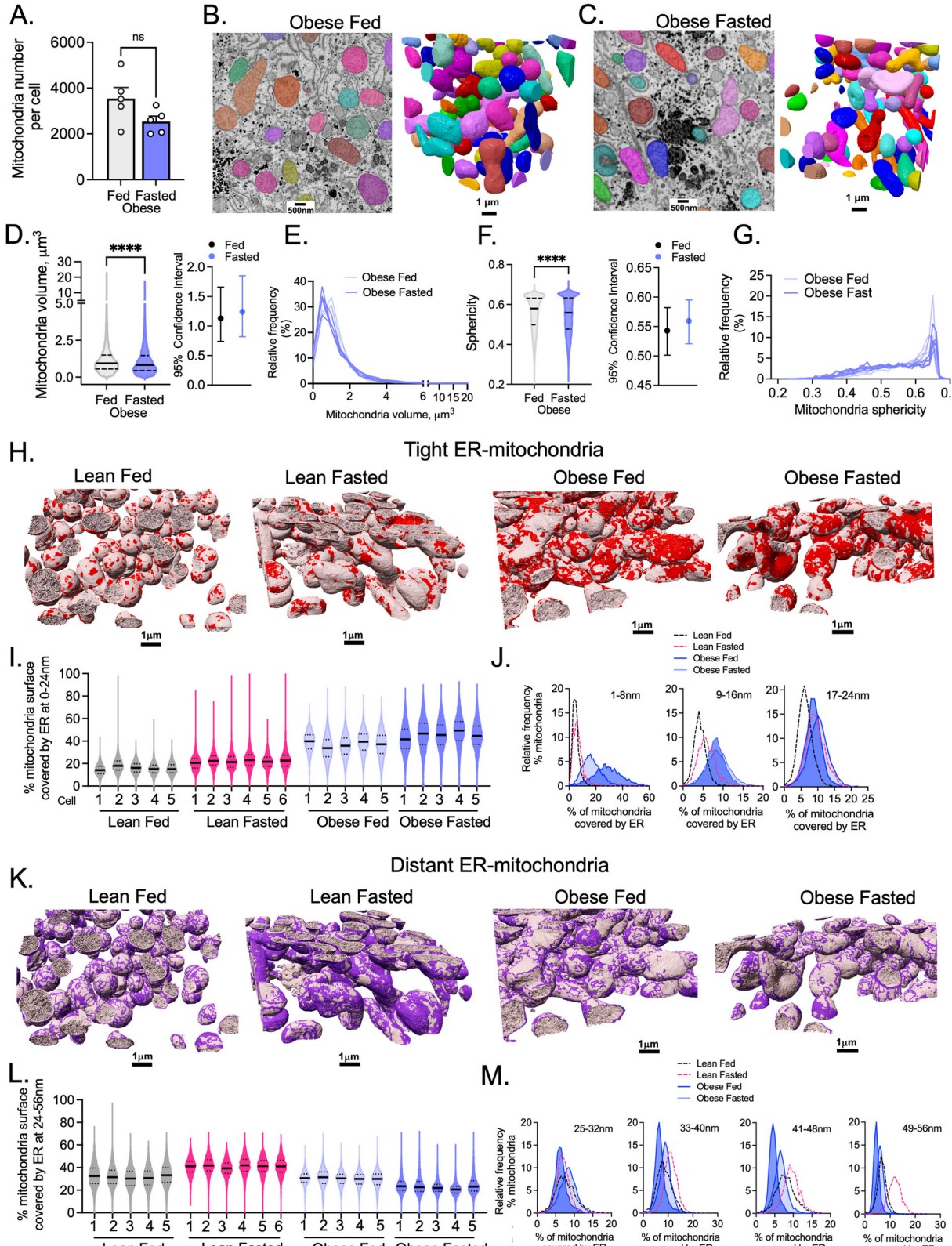

lobules aligns with areas where enhanced rough ER-mitochondria remodeling occurs during fasting.

To further investigate the endogenous localization of RRBP1 within the ER, we stained RRBP1 in primary hepatocytes isolated from fed and fasted mice and performed super-resolution fluorescence imaging, including lattice structured illumination microscopy (SIM) and stimulated emission depletion (STED) microscopy. As depicted in

Fig. 5D–F, in fasted mice, RRBP1 signal intensity is markedly higher near the mitochondria, a finding consistently observed with both lattice SIM (Fig. 5D) and STED (Fig. 5F). These observations suggest that RRBP1 levels are not only higher in fasting but also appear to re-localize to mitochondria-proximal regions.

Lastly, to test whether elevated levels of RRBP1 lead to the formation of rough ER -mitochondria interactions in hepatocytes, we

**Fig. 4 | Impact of obesity on fasting-feeding dynamics of ER, mitochondria, and their interactions in liver tissue. A** Number of mitochondria in individual hepatocytes from mice in fed and fasted state ($n = 5$ cells in each condition). **B** Left: SEM image (scale bar: 500 nm); Right: 3D segmentation and reconstruction of individual mitochondria in fed state (scale bar: 1 μm). **C** Left: SEM image (scale bar: 500 nm); Right: 3D segmentation and reconstruction of individual mitochondria in fasted state (scale bar: 1 μm). **D** Left: Mitochondria volume; Right: 95% of confidence interval for data in **D. E** Mitochondria volume frequency distribution. **F** Left: Mitochondria sphericity; Right: 95% of confidence interval for data in **F. G** Mitochondria sphericity frequency distribution. **H** 3D Reconstruction of mitochondria volumes marked by areas of interactions with the ER at 0–24 nm distance (red) lean fed, lean fasted, obese fed and obese fasted state conditions. Scale bars: 1 μm. **I** Quantification of the % of mitochondria surface covered by ER at 0–24 nm distance in the same conditions described in **H. J** Frequency distribution of % of mitochondria covered by ER at distances indicated in the graphs. **K** 3D

Reconstruction of mitochondria volumes marked by areas of interactions with the ER at 24-56 nm distance (purple) lean fed, lean fasted, obese fed and obese fasted state conditions. Scale bars: 1 μm. **L** Quantification of the % of mitochondria surface covered by ER at 24-56 nm distance in the same conditions described in **K. M** Frequency distribution of % of mitochondria covered by ER at distances indicated in the graphs. For (**D–G**) number of mitochondria are fed, $n = 17,721$ mitochondria and fasted, $n = 12,712$ mitochondria. For (**I, L**), lean fed ($n = 3173, 3000, 3109, 2787, 2787$), lean fasted ($n = 894, 1183, 1275, 1246, 757, 1323$), obese fed ($n = 2084, 5054, 3854, 3521, 3201$) and obese fasted ($n = 3250, 2781, 2607, 2065, 1999$) states. For (J, M), lean fed ($n = 2766$), lean fasted ($n = 1275$), obese fed ($n = 3851$), obese fasted ($n = 2586$) states. Obese: leptin-deficient mice, ob/ob. Fasted state corresponds to 20 h (overnight) food withdrawal. For the bar graph in **A**, data is shown as mean ± s.e.m; For **D** and **F**, two-tailed, unpaired $t$-test ****$p < 0.0001$ and permutation test ****$p < 0.0001$.

exogenously expressed RRBP1 in primary hepatocytes using an adenovirus-based system. As illustrated in Fig. 5G, RRBP1 gain of function results in significant remodeling of the ER in proximity with the mitochondria, which was not observed when we overexpressed SEC61β, another ER protein enriched in ER sheets.

## Lack of RRBP1 disrupts rough-ER mitochondria remodeling in fasting and obesity

The data presented above suggests a strong correlation between RRBP1 expression, localization and rough-ER-mitochondria interactions. To investigate the requirement of RRBP1 for fasting-induced ER sheet-mitochondria interactions, we examined organelle morphology and remodeling in mice lacking RRBP1 (RRBP1-KO). Using this in vivo model, we detected that the absence of RRBP1 in the liver significantly reduces ER-sheet-mitochondria interactions (25-80 nm) induced by fasting (Fig. 6A–C and Supplementary Fig. 10D). However, RRBP1-KO hepatocytes still maintain a high degree of peripheral ER sheets (Supplementary Fig. 10E). More interestingly, the absence of RRBP1 in livers leads to a marked alteration in the morphology of the mitochondria (Fig. 6B). Specifically, mitochondria in RRBP1-KO hepatocytes exhibit a rounder and swollen shape with a significant decrease in aspect ratio when compared to those in wild-type mice (Fig. 6D–F). To determine the impact of loss of RRBP1 on mitochondrial function, we measured mitochondria fatty acid oxidation capacity. For that, we loaded primary hepatocytes with $^{14}$C-labeled palmitate and measured the resulting $^{14}CO_2$ released through beta-oxidation. As shown in Fig. 6G, absence of RRBP1 markedly decreased the rate of mitochondria fatty acid oxidation. Accordingly, primary hepatocytes from RRBP1-KO mice exhibited significantly higher and bigger lipid droplet accumulation when loaded with oleic acid, compared to wild-type cells (Fig. 6H, I). Importantly, the increase in lipid droplet accumulation in RRBP1-KO cells is rescued by exogenous replenishment of RRBP1 in the cells both in the absence and presence of oleic acid (Fig. 6J, K)). Thus, lack of RRBP1 inhibits the formation of rough ER-mitochondria interactions induced by fasting and leads to defects in mitochondrial structure and function.

In the last part of our work, we tested whether decreased rough ER-mitochondria interactions in obesity could be related to alterations in RRBP1 abundance. For that, we determined the expression level of RRBP1 in liver tissues from obese mice in fed and fasted condition. As shown in Fig. 7A, RRBP1 expression is markedly reduced in livers from obese mice both in fed and fasted state. Next, we asked whether gain of function of RRBP1 in tissues of obese mice could per se rescue rough-ER mitochondria interactions. For that, we used adenovirus-mediated gene delivery to induce the expression of RRBP1 specifically in the liver of obese mice. Adenovirus expressing LacZ was used as a control (Fig. 7B). After performing both TEM and FIB-SEM analysis and automated image segmentation, we detected that gain of function of RRBP1 in the livers of obese mice significantly induced rough

ER-mitochondria interactions, at 25-56 nm distance (Fig. 7C–G). Interestingly, overall, the mitochondria shape in the hepatocytes overexpressing RRBP1 are flatter and more elongated (less sphericity) compared to obese hepatocytes expressing LacZ (Fig. 7 C, D, H, I), although total mitochondria volume is similar (Fig. 7J, K). Thus, RRBP1 is sufficient to restore fasting-induced ER sheet-mitochondria interactions in obese animals. Also, this data suggests that RRBP1 driven ER sheet-mitochondria interaction is important to regulate mitochondria shape. Taken together, these data show that RRBP1 is required for ER sheet remodeling in fasting and suggest that this subcellular structural organization is important for metabolic adaptation to fasting by regulating fatty acid oxidation.

## Discussion

Metabolic adaptation is essential to maintain homeostasis and survival during transitions in nutrient availability or metabolic stress. Hepatocyte metabolic adaptation to fasting and feeding has been shown to rely on changes in signaling pathways and transcription events[8]. However, metabolic processes are compartmentalized into subcellular domains and execution of complex metabolic programs requires dynamic organization of the cytoplasm and efficient communication of the organelles[2,43–45]. Only more recently, it's being recognized that metabolic adaptation also encompasses remodeling of subcellular architectural organization[2,10,46]. In this context, it's possible to envision that remodeling the shape of a single organelle, such as the ER, may support optimization of a given metabolic function in a given state. However, since the organelles do not exist in physical isolation in the cell, changes in structure of one may strongly impact others as well. Hence, a more holistic resolution of the complex subcellular architecture is critical for documenting the structural alterations induced by environmental factors and how they relate to functional outcomes. While this has been a formidable challenge due to technical limitations, recent developments in volume EM technology, which can capture structures at few nanometer-resolution in 3D, have opened the path to study tissue subcellular structure-function relationship[10].

Here, by using enhanced FIB-SEM and deep learning-based segmentation analysis combined with TEM and high-resolution fluorescence imaging, we showed that transitions from fed to fasted state induces a coordinated remodeling of ER and mitochondria shape and the way these organelles interact in 3D space. In fasting, mitochondria become larger and flatter likely due to increased fusion process. This is in accordance with previous observations in cell lines and liver tissue[47–49]. Increased mitochondria fusion has been shown to increase metabolic efficiency of oxidative phosphorylation and beta oxidation, and potentially protects mitochondria from autophagy[48,50].

In addition to mitochondrial remodeling, during fasting the rough ER sheets reorganize themselves to form a scaffold, surrounding these large mitochondria. These processes seem to be interrelated since deletion of RRBP1, which is required for ER sheet remodeling towards

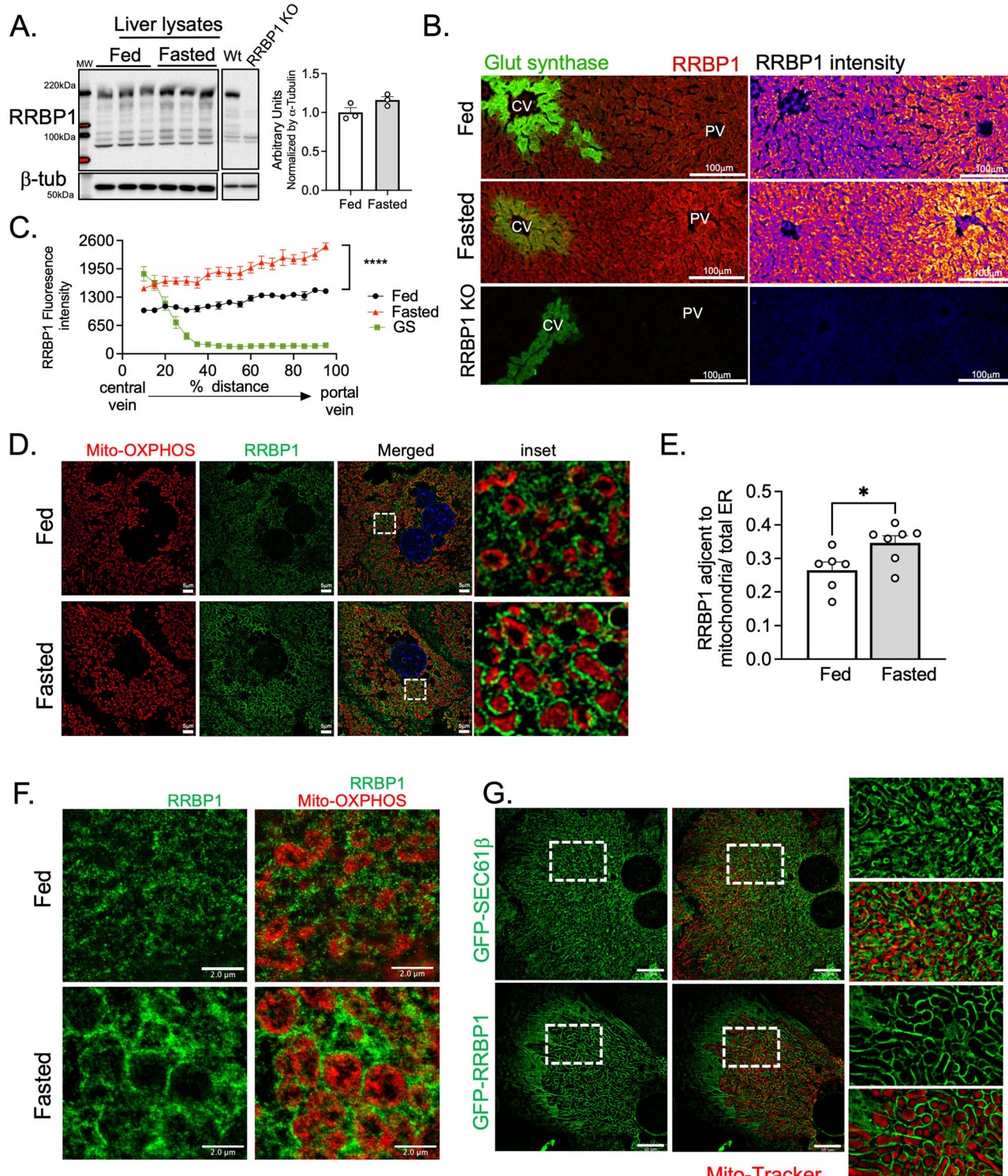

**Fig. 5 | RRBP1 expression in liver and primary hepatocytes in fed and fasting conditions. A** Immunoblotting and quantification analysis of RRBP1 protein in total liver lysates of indicated conditions. $n = 3$ in each group. **B** Immunostaining of RRBB1 and Glutamine synthetase (GS) in liver sections from lean mice in fed and fasted and from RRBP1 deficient mice. Left panel shows images pseudo colored by 'Fire' LUT. Scale bars: 100 μm. **C** Quantification of the immuno-stained tissue sections shown in B. For the quantification, a line was drawn from the edge of a central vein to the portal vein. The fluorescence intensity of RRBP1 and GS signals across each line was measured by plot profile and the distances were transformed into percentages. $n = 9$ fed, $n = 7$ fasted. Representative of 4 independent experiments. For C, two-way ANOVA, Sídák's multiple comparisons test ****$p < 0.0001$. **D** Immunostaining of RRBP1 (green) and mitochondrial OXPHOS proteins

(Oxphos cocktail antibody, red) in primary hepatocytes isolated from mice in fed and fasted state. Images were acquired with Lattice-SIM microscopy. Scale bars: 5 μm. **E** Quantification of RRBP1 fluorescence intensity in proximity with mitochondria. $n = 5$ cells fed and $n = 7$ cells fasted. Two-tailed, unpaired $t$ test, $p < 0.05$. **F** Immunostaining of RRBP1 (green) and mitochondrial OXPHOS proteins (red) in primary hepatocytes isolated from mice in fed and fasted state. Images were acquired with STED microscopy. Scale bars: 2 μm. Representative image from 7 cells per group. **G** Lattice-SIM fluorescence images of primary hepatocytes isolated from lean mice in fed state exogenously expressing Adenoviurs-GFP-Sec61β (upper panel) and adenovirus-GFP-RRBP1 (lower panel), representative of three independent experiments. Scale bars: 10 μm. Mitochondria were stained with MitoTracker. For the line graph in **C** and bar graph in **E**, data are shown as mean ± s.e.m.

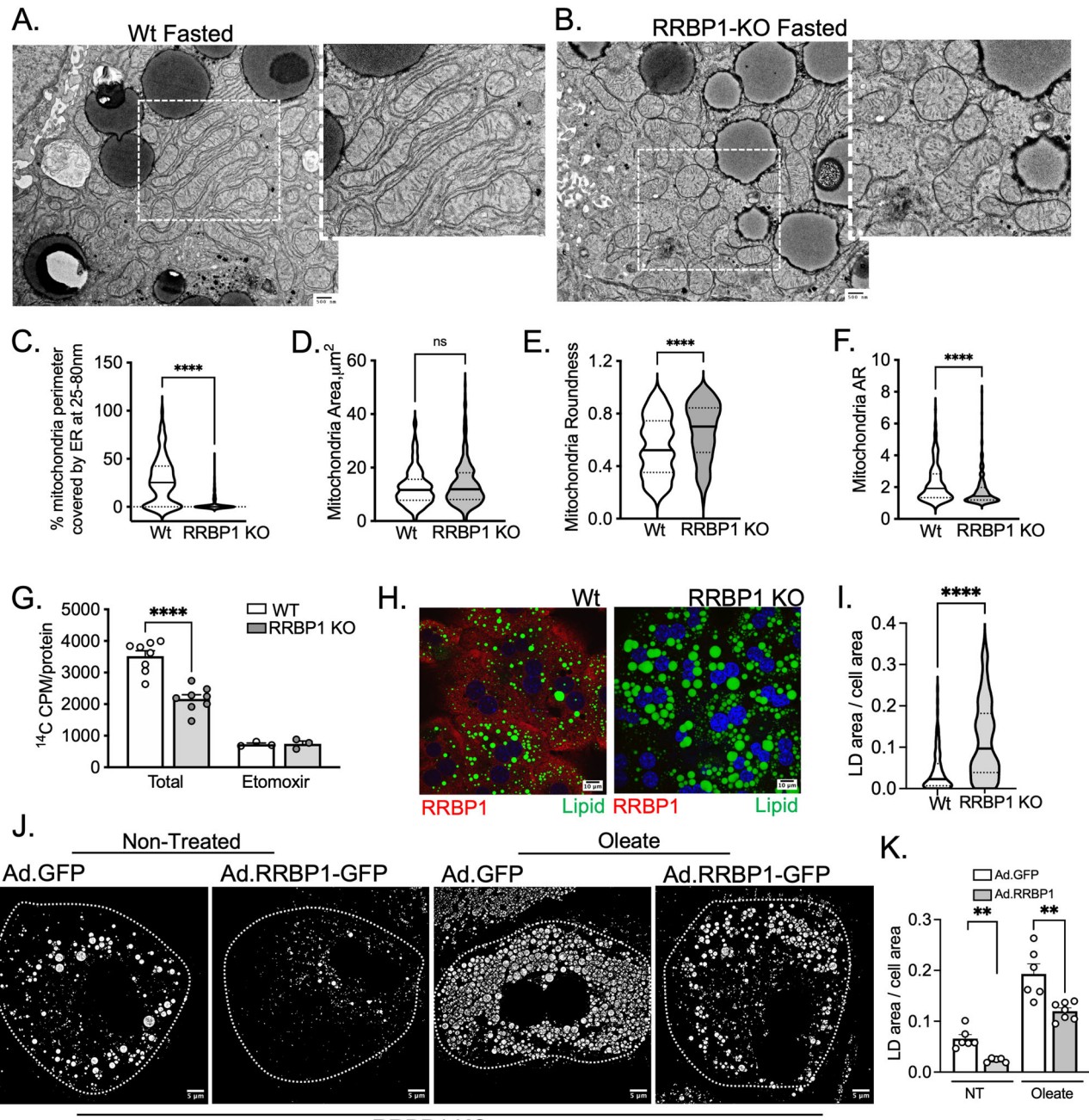

**Fig. 6 | Impact of hepatic RRBP1 deletion on ER-mitochondria interactions and mitochondria structure and function. A, B** Representative TEM images of livers from wild-type (WT) and RRBP1 deficient (RRBP1-KO) mice in fasted state. Scale bars: 500 nm. **C** Quantification of mitochondria surface covered by ER between 25-80 nm distance. $n = 208$ for Wt and $n = 311$ RRBP1-KO. **D–F** Quantification of mitochondria area (**D**), roundness (**E**) and aspect ratio (AR) (**F**), $n = 208$ for Wt and $n = 308$ RRBP1-KO. **G** $^{14}$C-palmitic acid-driven fatty acid oxidation in hepatocytes from wild-type control ($n = 8$) and RRBP1 KO ($n = 8$). **H, I** Oleate and palmitate driven lipid loading assay and quantification analysis in primary hepatocytes from wild type (Wt) and RRBP1 deficient cells ($n = 107$ cells for Wt, $n = 133$ cells for RRBP1 KO). Lipid droplets were stained with BODIPY (green) and RRBP1 was stained in red. Scale bars: 10 μm. **J, K** Oleate driven lipid loading assay in primary hepatocytes infected with adenovirus expressing GFP (ad-GFP) and adenovirus expressing RRBP1-GFP (ad-GFP-RRBP1), (GFP $n = 6$; RRBP1 NT $n = 5$, GFP Oleate $n = 6$, RRBP1 Oleate, $n = 7$). Scale bars: 5 μm. For the bar graphs data are shown as mean ± s.e.m.; For **C, E, F, G, I, K**, two-tailed unpaired t-test ****$p < 0.0001$, **$p < 0.003$.

the mitochondria, results in lack of ER sheet-mitochondria interactions and lack of mitochondria remodeling. Rather, mitochondria from RRBP1 deficient mice are swollen and assume a rounder morphology.

Interactions between rough ER and mitochondria have been observed in cell lines and recently described in both fed and fasted states in the liver, also referred to as the wrapper[20]. Our analysis also describes the presence of membrane organization consistent with these structures both in fed and fasted conditions. However, we show that in fasting, these interactions result from a global remodeling of ER

sheets—a process that depends on the protein RRBP1. Importantly, we show the remodeling that occurs during prolonged fasting, is preferentially located in mid-lobular and periportal zones and markedly downregulated in hepatocytes from obese mice.

The functional significance of rough ER sheet—mitochondria interactions are just starting to be explored. Rough ER—mitochondria interactions have been shown to be associated with the assembly of VLDL in the liver[20]. However, necessity of the rough ER's proximity to the mitochondria for this process remains unclear. Our experiments

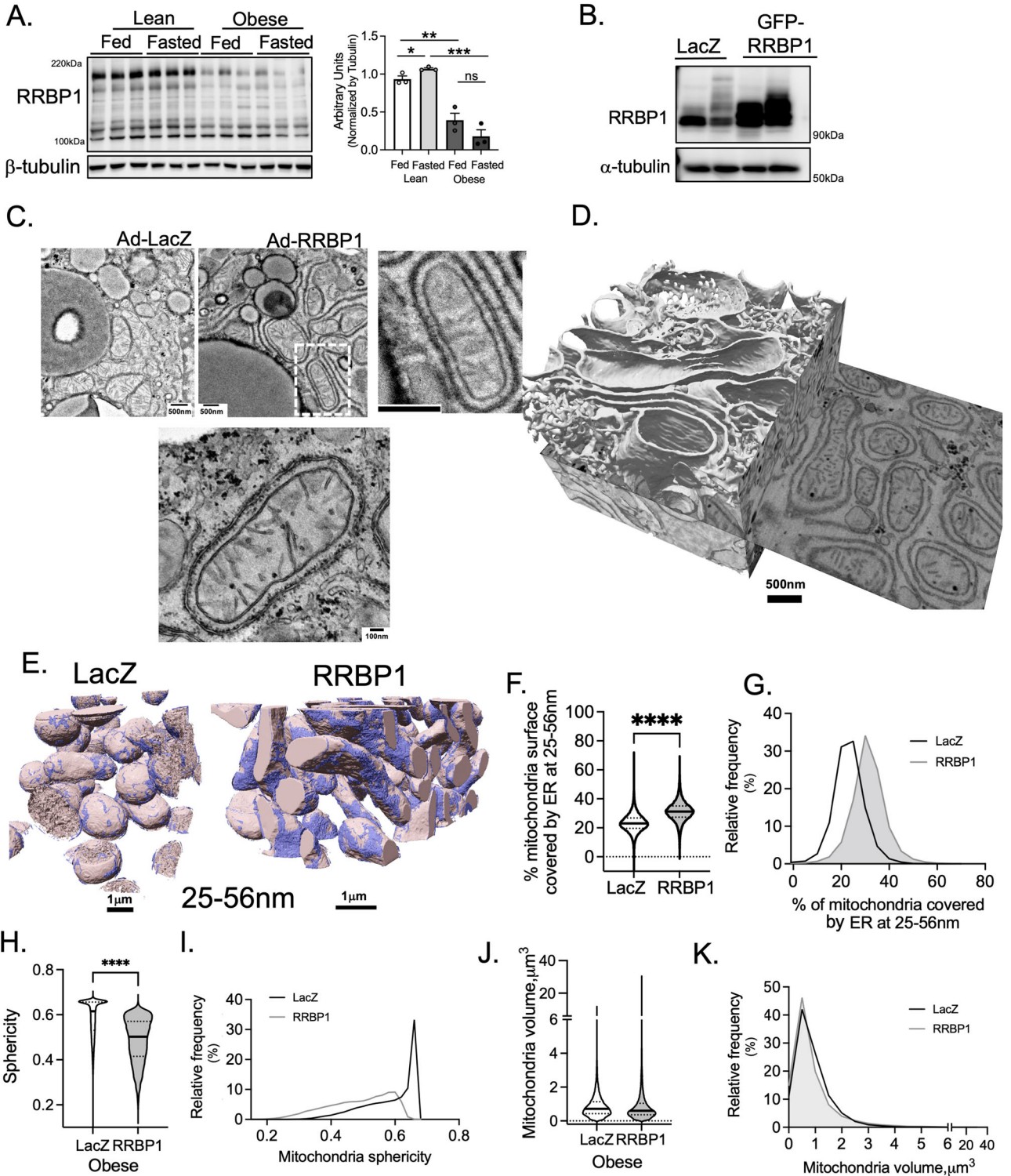

**Fig. 7 | Exogenous expression of RRBP1 rescues rough ER−mitochondria interactions in obesity. A** Immunoblotting and quantification analysis of RRBP1 protein in total liver lysates of indicated conditions. $n = 3$ in each group. **B** Immunoblot analysis of RRBP1 in total liver lysates from mice exogenously expressing LacZ ($n = 2$) or GFP-RRBP1 ($n = 2$) through Adenovirus gene delivery. **C** Representative TEM images from obese mice exogenously expressing LacZ or RRBP1 though adenovirus (Ad). Scale bars: 500 nm. **D** FIB-SEM imaging and 3D segmentation and rendering of liver volume from obese mice expressing RRBP1 depicting ER morphology. Scale bar: 500 nm. **E** FIB-SEM imaging and 3D segmentation and rendering of liver volume from obese mice expressing either LacZ (left) and RRBP1 (right), depicting mitochondria surface covered by ER at 25-56 nm distance. Scale bars: 1 μm. **F** Quantification of mitochondria surface covered by ER at 25-80 nm distance. **G** Mitochondria surface covered by ER at 25-80 nm distance frequency distribution. **H**, **I** Mitochondria sphericity and frequency distribution in liver tissue from LacZ and RRBP1 overexpression. **J**, **K** Mitochondria volume and frequency distribution in liver tissue from LacZ and RRBP1 overexpression. For **F**−**K**, $n = 23550$ mitochondria for LacZ and $n = 29235$ for RRBP1. Mitochondria were quantified in the entire data set imaged and not segregated by cell. For the bar graphs data are shown as mean ± s.e.m; For **F** and **H**, two-tailed unpaired $t$-test. ****$p < 0.0001$.

point to the direction that rough ER−mitochondria interaction is important to control mitochondrial fatty acid oxidation and overall mitochondria oxidative capacity. This is supported by the following evidence: a) genetic deletion of RRBP1, which disrupts rough ER−mitochondria interactions, results in decreased FAO and increased lipid droplet accumulation; b) RRBP1 deletion causes dysmorphic mitochondria and diminished elongation, which has been reported to as an important feature for efficient FAO[48]; c) fasting induced ER-mitochondria interactions are zonated towards mid-lobular and peri-portal regions, where the gradient availability of oxygen and mito-chondria efficiency are higher. Overall, it is clear that rough ER−mitochondria interactions play an important role in regulating liver lipid metabolism and further studies should expand this into addi-tional functional outcomes under different metabolic challenges or disease states.

The nature of ER-mitochondria interorganelle communication is highly complex and dynamic. Our study also illustrates the impor-tance of capturing the complexity of ER-mitochondria interactions extensively and holistically by using high-resolution 3-dimensional imaging and image analysis for clear understanding of the structural regulatory patterns of organelles. For example, we and several others have previously shown that increased MAMs in obesity lead to metabolic stress, mitochondrial dysfunction, and abnormal glucose homeostasis[9,18,36,51–54]. However, evidence for opposing conclusions have also been reported[37,38,55]. While such discrepancies may relate to the nature of the animals and their nutritional state, the analysis pre-sented here points to the critical value of precise resolution of struc-tures in high-volume tissue analysis. For example, a key factor is the ability to resolve and identify interaction of mitochondria with rough versus smooth ER and quantification of the nature of these interac-tions. While rough ER sheets can readily be seen in TEM due to the contrast of the ribosomes, identification of smooth tubular ER pre-sents significant challenges with TEM, hence limiting the ability to identify and quantify MAMs accurately. Another key determinant presented in this study is the fact that ER-mitochondria interactions are highly dynamic and differ according to liver zonation. Technically, it is possible that the methods of tissue preparation may also con-tribute to variation. All of our analysis of ER-mitochondria interactions was performed in tissues subjected to chemical fixation, which may affect the absolute measurements of ER-mitochondria distances. However, all samples examined were prepared with the same fixation method across all different conditions (fasting, feeding, obesity), which ensures accuracy in our comparative analysis.

An important finding of this study is that chronic stress induced by obesity leads to a marked impairment in dynamic organelle remo-deling as seen during cycles of fasting and feeding. In the obese state, the mitochondria stay in the round "fragment state" both in the fasting and feeding states. Second, the rough-ER-mitochondria interactions are not induced by fasting in obese hepatocytes. Third, obesity markedly induces tight smooth ER-mitochondria contacts (MAMs) both in fasted and fed states. The decrease in rough ER-mitochondria interactions in obesity could result from an overall decrease in the abundance of rough ER sheets, as we've previously shown, combined with the marked downregulation of RRBP1. In our previous work[10], we showed that overexpression of Climp-63 in the livers of obese mice leads to the proliferation of rough ER sheets and improved ER function and liver metabolism. Although we haven't explored the role of Climp-63 on fasting/feeding regulation, it is possible that rescue of total ER sheets can also lead to improved rough ER−mitochondria commu-nication and function in fasting.

On the other hand, the increase of MAMs could result from the proliferation of ER tubules in addition to the increased expression of tethering proteins and proteins that facilitate these contacts such as IP3Rs and others. Regardless, the obese state represents a loss of hepatocyte architectural flexibility which is consistent with the lack of

metabolic adaptive flexibility in this state. These findings reinforce the concept that architectural regulation is a critical determinant of metabolic homeostasis, which may have implications beyond the liver and relate to the regulation of many other functions.

## Methods

### General animal care, study design, and animal models
All animal experimentation was approved by the Institutional Animal Care and Use Committee at the Harvard T.H. Chan School of Public Health and University of California, Berkeley. The mice were housed at room temperature (70 °F) and at 30-70% relative humidity, on a 12 h light/dark cycle with free access to water and chow diet (PicoLab Mouse Diet 20 no. 5058, LabDiet) in the Harvard T.H. Chan School of Public Health pathogen-free barrier facility and in University of Cali-fornia, Berkeley pathogen-free barrier facility. The temperature and humidity control was monitored by operation personnel. We used the leptin-deficient B6.Cg-Lepob/J (ob/ob) male mouse (stock no. 000632) as a model of genetic obesity and aged/gender matched C57BL/6 J mice (Jackson Labs, stock no. 000664) or ob/+ heterozygotes (Jackson Labs, stock no. 000632) as lean controls. These animals were pur-chased around 6 weeks of age and used for experimentation between 9–11 weeks of age. RRBP1 heterozygous mice were obtained from the Mutant Mouse Resource and Research Center (MMRRC), NIH (stock no. 051080-JAX). Wild-type and homozygous RRBP1 null mice were obtained by crossing heterozygous mice at the Harvard T.H. Chan School facility and University of California, Berkeley facility. Mice were euthanized with $CO_2$ for tissue collection and biochemical analysis. For electron microscopy experiments, euthanasia was achieved by first, anesthetizing the mice with 300 mg kg$^{-1}$ ketamine and 30 mg kg$^{-1}$ xylazine in PBS, followed by perfusion with 10 ml saline and then by 10 ml fixative buffer containing 2.5% glutaraldehyde, 2.5% paraf-ormaldehyde in 0.1 M sodium cacodylate buffer (pH 7.4) (Electron Microscopy Sciences, catalog no. 15949). Number of replicates and sample size were determined based on prior research[56,57] and all measurements were included in the analysis. All samples were pro-cessed using the same methods. Imaging studies could not be done blinded because of the evident intrinsic features of the samples. For the time-course studies, tissues were collected at 9 a.m. (ad-libitum fed mice) and after 6 h (3 p.m.), 10 h (7 p.m.), 16 h (1 a.m.), and 24 h (9 a.m. next day) of food withdrawal. Mice were maintained in cages without food. In the imaging studies, the fed state corresponds to 6 h (9 a.m. − 3 p.m.) food withdrawal and the fasted state corresponds to 20 h (6 p.m. − 2 p.m.) overnight food withdrawal.

### Adenovirus-mediated overexpression of RRBP1
EGFP tagged human RRBP1 (p180) construct was a gift from Tom Rapoport's lab at Harvard Medical School. For in vivo exogenous expression, RRBP1 construct was cloned in an adenovirus (Ad5) backbone from Vector BioLabs. The adenovirus (serotype 5, Ad5) was generated, amplified, and double purified in CsCl by Vector BioLabs and it was administered to 8-week-old ob/ob mice, at a titer of $1 \times 10^9$ infectious units (IFU) per mouse. The mice were euthanized, and tis-sues collected 10-14 days after adenovirus injection.

### Fatty acid oxidation and lipid loading assays
Fatty acid oxidation (FAO) assay was performed as described previously[10,58]. Briefly, primary hepatocytes isolated from indicated mice were incubated in Media-199 with 25 mM HEPES, 1% BSA fraction V, and 100 nM glucagon overnight, in the absence of serum. The next day, 250 µM sodium palmitate was added to the cells for 2 h, followed by a spike with 0.5 µCi per well of $^{14}$C-palmitic acid (Perkin Elmer, catalog no. NEC075H250UC) and incubated for another 90 min. $^{14}$C-labeled $CO_2$ gas was captured as a result of FAO as described and radioactivity was measured. Etomoxir (40 µM), an inhibitor of the carnitine palmitoyl transferase I, was used in some wells to inhibit FAO,

30 min prior to palmitate addition. For lipid loading experiments, primary hepatocytes were treated with 1 mM oleic acid and 40 μM palmitic acid in Williams Media for 16 h and stained with BODIPY (green) (or AutoDOT (blue), (Abcepta, catalog: SM1000a) if the cell was expressing GFP) and Hoechst to visualize lipid droplets and nucleus, respectively. For the adenovirus rescue and imaging experiments, cells were incubated with either GFP-RRBP1, GFP-SEC61β or GFP adenovirus overnight, at 40 MOI dose.

## Total protein extraction and immunoblotting

Frozen liver tissues were homogenized in polytron device in cold lysis buffer containing 50 mM Tris-HCl (pH 7.4), 2 mM EGTA, 5 mM EDTA, 30 mM NaF, 10 mM $Na_3VO_4$, 10 mM $Na_4P_2O_7$, 40 mM glycerophosphate, 1% NP-40 and freshly added protease inhibitor cocktail (1%). Cell debris was discarded after centrifuging the samples at 9000 rpm for 15 min and protein concentrations were equalized using a BCA assay. Samples were heated at 95 °C for 5 min in Laemmli buffer containing 2-Mercaptoethanol and subjected to SDS–polyacrylamide gel electrophoresis. Proteins in the gel were transferred to a PVDF membrane and blocked with 5% BSA as previously described[10]. Membranes were then incubated with primary antibody overnight at 4 °C and with secondary antibody conjugated with horseradish peroxidase (HRP) for 2 h at room temperature, next day. Western blots were visualized using the SuperSignal West Pico Plus Chemiluminescent Substrate (Thermo Scientific). The uncropped versions of the western blots are provided together with the source data. Subcellular fractions enriched in endoplasmic reticulum vesicles were obtained by differential centrifugation following the same protocol as described in detail[10].

## Primary and secondary antibodies

RRBP1 antibody was obtained from Proteintech (catalog: 22015-1-AP) and validated with knockout controls as shown in Fig. 5A. Calnexin antibody was obtained from Santa Cruz Biotechnologies (catalog: sc-6465), alpha-tubulin was obtained from Proteintech (catalog: 66031-1-Ig) and beta-tubulin was obtained from Abcam (catalog: ab21058). HRP-conjugated secondary antibodies were obtained from Cell Signaling (anti-rabbit, catalog: 7074) and Santa Cruz Biotechnologies (anti-goat, catalog: sc-6465). For western blotting, primary antibodies were used at 1:1,000 dilution and secondary HRP-conjugated antibodies were used at 1:5,000 dilution. For immunofluorescence, glutamine synthetase (Santa Cruz Biotechnology, 74430), OXPHOS (Abcam, ab110413), STED Abberior STAR-Orange and STAR-Red secondary antibodies were used.

## EM sample preparation and TEM image acquisition

TEM and FIB-SEM sample preparations were done as previously described[10]. Briefly, mice livers were perfused via portal vein with saline followed by a fixative containing 2.5% glutaraldehyde and 2.5% paraformaldehyde in 0.1 M sodium cacodylate buffer (pH 7.4) (Electron Microscopy Sciences, catalog no. 15949). Tissues were cut in 300-micron thick slices with a Compresstome and were further incubated in a fresh fixative containing 1.25% formaldehyde, 2.5% glutaraldehyde, 0.03% picric acid, 0.05 M cacodylate buffer. For TEM imaging, samples were embedded in resin and ultrathin sections were prepared using a Reichert Ultracut-S microtome and imaged with a JEOL 1200EX electron microscope at 80 kV. For FIB-SEM, samples were prepared following the rOTO protocol as described previously in detail[10] and embedded in Durcupan resin.

## FIB-SEM sample preparation

A total six samples of various metabolic conditions (Lean fasted, Obese fasted, Lean peri-central, Lean peri-portal, Obese ad.LacZ and Obese ad.RRBP1) were prepared for FIB-SEM imaging as described[59]. Specifically, each sample was first mounted to the top of a 1 mm copper post which was in contact with the metal-stained sample for belter charge

dissipation. The vertical sample posts were each trimmed to a small block containing Region of Interest (ROI) with a width perpendicular to the ion beam, and a depth in the direction of the ion beam. The block sizes were roughly $100 \times 100 \ \mu m^2$. The trimming was guided by X-ray tomography data obtained by a Zeiss Versa XRM-510 and optical inspection under a Leica UC7 ultramicrotome. Thin layers of conductive material of 10-nm gold followed by 100-nm carbon were coated on the trimmed samples using a Gatan 682 High-Resolution Ion Beam Coater. The coating parameters were 6 keV, 200 nA on both argon gas plasma sources, and 10 rpm sample rotation with 45-degree tilt.

## FIB-SEM 3D large volume imaging

These six FIB-SEM prepared samples were imaged by two customized Zeiss FIB-SEM systems previously described[60,61]. Each block face of ROI was imaged by an electron beam with 1.2 keV landing energy at 2 MHz. A 3-nA imaging current was used for the Lean fasted and Obese fasted samples, while a 2-nA imaging current was for the Obese LacZ, Obese RRBP1, peri-central, and peri-portal samples. The x-y pixel resolution was set at 8 nm. A subsequently applied focused Ga+ beam of 15 nA at 30 keV strafed across the top surface and ablated away 8 nm of the surface. The newly exposed surface was then imaged again. The ablation−imaging cycle continued about once every 35−92 seconds for 4−10 days to complete the six samples. Each acquired image stack formed a raw imaged volume, followed by post processing of image registration and pairwise alignment using a Scale Invariant Feature Transform (SIFT) based algorithm. The aligned stacks consist of a final isotropic volume of $64 \times 56 \times 76 \ \mu m^3$, $75 \times 55 \times 68 \ \mu m^3$, $52 \times 52 \times 62 \ \mu m^3$, $75 \times 85 \times 47 \ \mu m^3$, $65 \times 65 \times 86 \ \mu m^3$, and $52 \times 52 \times 72 \ \mu m^3$ for Lean fasted, Obese fasted, Obese LacZ, Obese RRBP1, peri-central, and peri-portal, respectively.

## FIB-SEM segmentation and data analysis

ER, mitochondria, lipid droplets, nucleus and cell borders were segmented using the 3dEMtrace platform of ariadne.ai (https://ariadne.ai/) by using a customized convolutional neural network (CNN) architecture based on 3D U-Net. Binary tiff masks (for ER and nucleus) and instance-based segmentation (for mitochondria, LD and cell borders) were generated, to assign a unique identifier to each organelle. The raw data (from Lean-Fed and Obese-Fed datasets) used to extract mitochondria instance segmentation were published previously[10]. The mitochondria data from these two and other new datasets are extracted and re-segmented individually and used to quantify inter-organelle interactions. The false positive connections between different mitochondria were improved by the instance-based segmentation. Data were analyzed in Arivis Vision 4D (Zeiss) software by creating objects from segmented data. After generating the 3D objects (full connectivity in x, y, z), proof-reading was done to eliminate the unspecific ER objects less than 100,000 voxels in all datasets. Minimum mitochondria object voxel volumes are as follows per dataset: Lean Fed: 21696 Lean Fasted: 92243, Obese Fed: 50382, Obese Fasted: 80293, Obese LacZ: 50437, Obese RRBP1: 50234. Organelles were then re-segmented by the cell that they belong (5 hepatocyte volumes per condition).

**Quantification of ER-Mitochondria and ER-Lipid droplet interactions.** For the quantification of ER-mitochondria or lipid droplet-mitochondria interactions, single-voxel-thick mitochondria outer surface was generated by eroding the mitochondria volume (plane-wise) and subtracting this eroded volume from the original volume. Then, the interacting organelle (ER or LD) annotated voxels were expanded (plane-wise) at various distances (24 or 56 nm) and the overlapping voxels of the expanded volume and mitochondria outer surface were marked as "interaction surface". To calculate the percentage of mitochondria surface interacting with the corresponding organelle, the

interaction surface was divided by the single-voxel-thick mitochondria outer surface.

Voxel-by-voxel ER-mitochondria interaction was calculated by dilating ER annotated voxels one at a time. After getting the intersection with the outer mitochondria voxel, these overlapped voxels were deleted to mark the consecutive step (to avoid counting the same voxel multiple times).

ER sheet−mitochondria interaction was calculated similarly as described above, by using the ER sheet segmentation as described previously[10].

**Spatial distribution of mitochondria within the hepatocyte.** The spatial distribution of mitochondria within hepatocytes was analyzed by measuring distances from the center of the mitochondria's 3D geometries to the cell nucleus. For hepatocytes containing two nuclei, distances were measured to the midpoint between the nuclei to maintain consistency. These measurements were visualized using color-coded representations in Arivis Vision 4D software. Distances were categorized into absolute distances, with colors ranging from blue (indicating proximity to the nucleus) to red (representing distances of 25 μm or greater from the nucleus). Similarly, relative distances were visualized, with blue marking the closest to nucleus and red indicating the furthest (independent of cell size), to provide insights into mitochondrial positioning within the cell.

Mitochondria complexity index was calculated by the following formula described in ref. [11]: $MCI^2 = SA^3 / (16 \pi^2 V^2)$, where SA is surface area in $\mu m^2$ and $V$ is volume in $\mu m^3$.

### TEM data analysis

For 2D TEM analysis, mitochondria and ER borders were manually drawn using the ROI Group Manager in FiJi. The distance and surface coverage calculations were done using a macro, provided at https://github.com/SabriUlkerCenter. In summary, the ER mitochondria distance was calculated by generating a distance map from the mitochondria perimeter and assigning pixel intensity values to the ER channel based on its distance to the corresponding mitochondrion. The parallel ER sheets were calculated by the Matlab algorithm as previously described[10].

### RRBP1 immunofluorescence and zonation

Liver sections from fed, fasted (20 h) and RRBP1-KO (fed) mice were cut into 8.0 mm slices using Disposable Skin Biopsy Punches (Electron Microscopy Science, Hatfield, PA, USA), blotted dry, and placed in a cryo-mold on a thin layer of Optimal Cutting Temperature (OCT) embedding media (Fisher Healthcare, Houston, TX, USA) and then covered completely with OCT. OCT-embedded samples were gradually frozen in liquid nitrogen vapor and further stored at −80 °C until sectioned. Cryosections (5 μm) were prepared using the Leica CM3050S Cryotome (Leica Biosystems, Wetzlar, Germany). Sections were fixed with 4% paraformaldehyde (Electron Microscopy Sciences) for 10 min at room temperature (RT) and washed 3x in PBS, before a 20 min permeabilization using 0.2% Triton-X100 at RT. Sections were blocked with 5% BSA in PBS for 1 h and incubated with primary antibody solutions at 4 °C overnight. Rabbit anti-RRBP1 (#22015-1-AP, ProteinTech) antibody was used at a concentration of 1:100, while mouse anti-GS (#74430, Santa Cruz Biotechnology) was used at 1:200 in blocking buffer. The next day, sections were washed 3x, including one long wash for more than 10 min. Fluorochrome-coupled secondary antibodies were diluted 1:1000 in PBS, and the cells were incubated with them at RT for 2 h in the dark. The sections were washed 3x, including one long wash, and if needed, Hoechst was used as nuclear marker, diluted 1:1000 in PBS for 10 min at RT. Following three washing steps with PBS, sections were mounted with ProLong Gold antifade reagent (Thermo Fisher Scientific) and immunofluorescence analysis was performed using the LSM880 FCS confocal inverted

microscope (Zeiss, Oberkochen, Baden-Württemberg, Germany) using a 10x objective lens. ZEN Black software (Zeiss) was used for acquisition parameters, shutters, filter positions and focus control. Image analysis was performed using FiJi software. All the images in the same experiment were analyzed using the same parameters. In each image analyzed, a line was hand-drawn between the edge of a central vein and the portal vein. The fluorescence intensity of RRBP1 and GS signals across each line was measured by plot profile. The distance (in micrometers) between central vein and portal vein were transformed into percentages, where 0% and 100% correspond to the edges of a central and portal vein, respectively. The percentages were binned in a range of 5, and the fluorescence intensity was presented as the average for each bin value. For statistical analysis, we used the fluorescence intensity average for the bins 10 and 90 to represent the central and portal vein, respectively, for 4 independent experiments.

### RRBP1 fluorescence imaging

To evaluate the endogenous localization of RRBP1 protein, primary hepatocytes were isolated from lean mice in fed and fasted (overnight-16h) conditions. Next day, cells were fixed with 4% paraformaldehyde for 10 min at RT and washed with PBS 3 times, followed by a 20 min permeabilization using 0.2% Triton-X100 at RT. RRBP1 protein was stained with Rabbit anti-RRBP1 (#22015-1-AP, ProteinTech) antibody overnight at +4 °C, at a concentration of 1:100 in PBS. Mitochondria was stained with total OXPHOS Rodent WB Antibody Cocktail (Abcam, ab110413) at 1:200 dilution. For Lattice-SIM (Elyra7, Zeiss) super-resolution microscope imaging, cells were stained with Alexa Fluor secondary antibodies at 1:1000 dilution. For stimulated emission depletion (STED) microscopy, Abberior secondary dyes (STAR orange and red) were used at 1:1000 dilution. Lattice-SIM images were acquired on a Zeiss Elyra7 super-resolution fluorescence microscope, equipped with dual sCMOS PCO Edge 4.2 cameras for simultaneous two channel acquisition, with a 63x/1.4 oil objective. For each focal plane 13 phase images were acquired. Lattice-SIM reconstruction was performed with the SIM processing Tool of the ZEN 3.0 SR (Black, v16) software. Stimulated emission depletion (STED) super-resolution imaging was carried out on the STEDYCON, an Abberior STED system coupled to a Nikon Ti2 body via the C-port and by using Nikon CFI Plan Apo Lambda 100X/1.45NA oil objective. Samples were fixed as described above. RRBP1 was detected using STED Abberior STAR-Red dye and mitochondria was detected by using the OXPHOS antibody with STAR-Orange secondary dye. The 775-nm depletion lasers were used.

### Statistics and reproducibility

Statistical significance was assessed using GraphPad Prism version 9, by the unpaired *t*-test (two-tailed) or One-way Anova analysis and *P* values are indicated in the figure legends. For in vitro studies a minimum of three biological replicates were used for each experiment. For FIB-SEM data, we also calculated the 95% intervals, presented in each figure. In the statistical analysis, permutation tests (with *n* = 1e5) were employed to assess differences between conditions by pooling cells within each weight (lean/obese) and feeding (fed/fasted) group. This approach generates an empirical p-value by calculating the proportion of instances where the original sample's metric exceeds that of the permuted samples. To complement the permutation tests and illustrate effect size, we performed bootstrap resampling (*n* = 20) to generate 95% confidence intervals for the sample (Supplementary Fig. 11). This dual methodological framework, combining permutation tests with bootstrapping, enables a comprehensive evaluation of the statistical significance and confidence in the observed differences.

### Reporting summary

Further information on research design is available in the Nature Portfolio Reporting Summary linked to this article.

## Data availability

The previously published[10] raw FIB-SEM volumes of the lean and obese mice in the fed state can be reached at https://doi.org/10.6019/EMPIAR-10791. The different features that were extracted and analyzed in this paper, such as instance-based mitochondria segmentation and other raw FIB-SEM (Lean fasted, Obese fasted, Obese LacZ, Obese RRBP1, pericentral, peri-portal) and segmentation data deposited to the EMPIAR[62] database and can be reached here: https://doi.org/10.6019/EMPIAR-12017. Supplementary videos can also be reached at https://www.youtube.com/playlist?list=PLpzquMkvsJ9WCAl5Uea2W1vzgkHjjxfWL. The uncropped raw versions of the western blots generated during the current study are provided together with the source data. Source data are provided with this paper.

## Code availability

The analysis pipeline and code used to analyze the organelle interactions and quantifications in Supplementary Fig. 4F can be found at https://github.com/SabriUlkerCenter.

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

## Acknowledgements

We are very grateful to Harald F. Hess for initiating our collaboration with Janelia, HHMI. We thank E. Benecchi, M. Ericsson and L. Trakimas for their help in sample preparation for TEM; C. Zugates for his help and guidance in using the Arivis Vision 4D software; A. Wanner, J. Kornfeld, and the Ariadne team for their effort and help with the segmentation; Dr. Karsten Bahlmann for his assistance with STED microscopy, Dr. Brian Lawney for his help and guidance with the statistical analysis and all members of the Hotamisligil laboratory community for their continued support and encouragement. We thank H. M. Leung and R. Anadol for their artistic talent and vision, visualization of the data, generating videos, and sharing the resources and expertize of the Refik Anadol Studio. This project is supported by the Hotamisligil Lab and Sabri Ülker Center for Metabolic Research. G.P. is supported by an NIH training grant (5T32DK007529-32). L.L.A received scholarship from Fundação de Amparo à Pesquisa do Estado de São Paulo (FAPESP, process 2019/04943-3 and 2022/04464-0). S.P. and C.S.X. were supported by HHMI. Dr. Arruda is supported by the Chan-Zuckerberg BioHub and by NIH 5P30DK116074.

## Author contributions

G.P. and A.P.A. formulated the questions, designed the project, and performed the in vitro and in vivo experiments, analyzed the data, prepared the figures and wrote the manuscript. E.C., N.M., L.L.A., R.V., R.L.S.G., and Y.L. performed and assisted with in vitro and in vivo experiments and some of the image analysis. S.P. and C.S.X. performed, supervised, and executed collection of the FIB-SEM data. G.S.H. and A.P.A. conceived and supervised the project, designed experiments, interpreted results, and revised the manuscript.

## Competing interests

C.S.X. is the inventor of a US patent assigned to HHMI for the enhanced FIB-SEM systems used in this work: Xu, C. S., Hayworth, K. J., and Hess H. F. Enhanced FIB-SEM systems for large-volume 3D imaging. US Patent 10,600,615 (2020). The other authors declare no competing interests.
