## [Peer Review File · Nature Communications]

Spatial mapping of hepatic ER and mitochondria architecture reveals zonated remodeling in fasting and obesityEditorial Note: This manuscript has been previously reviewed at another journal that is not operating a transparent peer review scheme. This document only contains reviewer comments and rebuttal letters for versions considered at *Nature Communications*.

REVIEWERS' COMMENTS

Reviewer #1 (Remarks to the Author):

In the revised manuscript entitled "Spatial mapping of subcellular liver architecture reveals zoned organelle remodeling in fasting and obesity" the authors addressed my concerns and also those of the other reviewers with great care. This resulted in the improved visualization of the existing data and the inclusion of new interesting data such as the mitochondrial distribution within whole liver cells under different metabolic conditions (Ext. Data Fig. 6). Also, the modified title better highlights the strength of the manuscript with an emphasis on zonation which was also found in the distribution of RRB-1.

Minor point:

Maybe I overlooked this, but it seems that Ext. Data Fig. 11 was not referenced anywhere in the text.

Taken together, the current manuscript is unique in terms of the large-scale 3D analysis. Such data can easily be dismissed as descriptive, although precisely such observations are a prerequisite for the development of new concepts. Such insight into mitochondrial and ER remodeling in reaction to metabolic states by investigating whole organelle populations in complete cells is novel and very valuable. This outweighs the inevitable low n-number, which is complemented by parallel TEM investigations on larger group sizes. Seen the substantial changes I am convinced that the manuscript is acceptable for publication.

Reviewer #2 (Remarks to the Author):

The revised study has largely addressed the previous concerns. New data further explore ER morphology and ER-LD contacts in the different nutrient conditions.

A major point in the revision was to examine how cristae change in feeding/fasting. While there are limitations to the EM because of limited volumetric measurements of the mitochondrial cristae, the data presented in the rebuttal letter Fig 1 appear relevant to the study as a whole. My only remaining suggestion is that this Fig 1 letter data should still be incorporated into the present study, rather than omitted.

Reviewer #3 (Remarks to the Author):

The authors have successfully addressed my concerns. I approve publication of this paper.

REVIEWERS' COMMENTS

Reviewer #1 (Remarks to the Author):

In the revised manuscript entitled "Spatial mapping of subcellular liver architecture reveals zoned organelle remodeling in fasting and obesity" the authors addressed my concerns and also those of the other reviewers with great care. This resulted in the improved visualization of the existing data and the inclusion of new interesting data such as the mitochondrial distribution within whole liver cells under different metabolic conditions (Ext. Data Fig. 6). Also, the modified title better highlights the strength of the manuscript with an emphasis on zonation which was also found in the distribution of RRBP-1.

Minor point:

Maybe I overlooked this, but it seems that Ext. Data Fig. 11 was not referenced anywhere in the text.

Taken together, the current manuscript is unique in terms of the large-scale 3D analysis. Such data can easily be dismissed as descriptive, although precisely such observations are a prerequisite for the development of new concepts. Such insight into mitochondrial and ER remodeling in reaction to metabolic states by investigating whole organelle populations in complete cells is novel and very valuable. This outweighs the inevitable low n-number, which is complemented by parallel TEM investigations on larger group sizes.

Seen the substantial changes I am convinced that the manuscript is acceptable for publication.

We thank the reviewer for their kind remarks and feedback during the revision process. The Supplementary Figure 11 is referenced in the Statistics and Reproducibility (Methods) section of the manuscript.

Reviewer #2 (Remarks to the Author):

The revised study has largely addressed the previous concerns. New data further explore ER morphology and ER-LD contacts in the different nutrient conditions.

A major point in the revision was to examine how cristae change in feeding/fasting. While there are limitations to the EM because of limited volumetric measurements of the mitochondrial cristae, the data presented in the rebuttal letter Fig 1 appear relevant to the study as a whole. My only remaining suggestion is that this Fig 1 letter data should still be incorporated into the present study, rather than omitted.

We thank the reviewer for their feedback. We value the reviewer's suggestion to incorporate the data from Fig 1 of our rebuttal letter. This data will be available in the Peer Review file. At this point, we prefer not to include it in the main manuscript due to the methodological limitations of accurately capturing cristae measurements. We are currently re-segmenting cristae to analyze its organization, abundance and other features in greater detail in a forthcoming manuscript. We believe this analysis is not the main focus of the current work. We appreciate the opportunity to clarify our approach and are grateful for the constructive dialogue throughout this review process.

Reviewer #3 (Remarks to the Author):

The authors have successfully addressed my concerns. I approve publication of this paper.

We thank the reviewer for their comments and feedback during the revision process.